# Permeabilization-free *en bloc* immunohistochemistry for correlative microscopy

Kara A Fulton[1,2,3], Kevin L Briggman[2,3]*

[1]Brown University, Providence, United States; [2]National Institute of Neurological Disorders and Stroke (NINDS), Bethesda, United States; [3]Center of Advanced European Studies and Research (caesar), Bonn, Germany

**Abstract** A dense reconstruction of neuronal synaptic connectivity typically requires high-resolution 3D electron microscopy (EM) data, but EM data alone lacks functional information about neurons and synapses. One approach to augment structural EM datasets is with the fluorescent immunohistochemical (IHC) localization of functionally relevant proteins. We describe a protocol that obviates the requirement of tissue permeabilization in thick tissue sections, a major impediment for correlative pre-embedding IHC and EM. We demonstrate the permeabilization-free labeling of neuronal cell types, intracellular enzymes, and synaptic proteins in tissue sections hundreds of microns thick in multiple brain regions from mice while simultaneously retaining the ultrastructural integrity of the tissue. Finally, we explore the utility of this protocol by performing proof-of-principle correlative experiments combining two-photon imaging of protein distributions and 3D EM.

## Introduction

Anatomical reconstructions of synaptic connectivity are one essential component of understanding neuronal circuits in the nervous system. Recent improvements in automating the collection of 3D electron microscopy (EM) data have dramatically increased the tissue volumes that can be acquired (*Briggman and Bock, 2012*; *Eberle et al., 2015*). Connectivity data derived from EM, however, does not alone contain sufficient information about the functional properties of neurons to fully constrain biologically plausible computational models. One approach to augment EM datasets with functional information is to physiologically record from neurons in a tissue sample prior to fixation (*Bock et al., 2011*; *Briggman et al., 2011*; *Lee et al., 2016*; *Wanner and Friedrich, 2020*). As a complementary approach, fluorescence immunohistochemistry (IHC) can be used to identify the constituent proteins of neuronal cell types and synapses that are key determinants of the functional properties of neurons (*Coons et al., 1942*; *Cuello et al., 1983*).

Post-sectioning IHC is typically employed for correlative EM because the sectioning process provides access to intracellular epitopes. For example, array tomography was developed to multiplex the labeling of numerous targets in ultrathin tissue sections (*Collman et al., 2015*; *Micheva and Smith, 2007*). However, this approach is incompatible with *en bloc*-based methods for collecting 3D EM data that require pre-embedding IHC, as in the case of focused ion beam scanning electron microscopy (FIB-SEM, *Hayworth et al., 2015*; *Knott et al., 2008*), gas cluster ion beam SEM (*Hayworth et al., 2020*), and serial block-face scanning electron microscopy (SBEM, *Denk and Horstmann, 2004*). For pre-embedding IHC, a membrane permeabilization step with nonionic surfactants such as Triton X-100 or polysorbate 20 (Tween 20) is typically performed to enable antibody penetration into thick tissue blocks, but at the cost of degraded tissue ultrastructure (*Helenius and Simons, 1975*; *Humbel et al., 1998*).

*For correspondence:
kevin.briggman@caesar.de

Competing interests: The authors declare that no competing interests exist.

To address this limitation, we began with the observation that the preservation of extracellular space (ECS) during chemical tissue fixation improves the penetration of antibodies into the mammalian retina using minimal permeabilization (*Pallotto et al., 2015*). Building on this finding, we developed a protocol that is capable of immunohistochemically labeling proteins throughout acutely fixed brain tissue sections up to 1 mm thick without permeabilization. Here, we describe the key changes that were made in comparison to conventional IHC protocols, leading to improved ultrastructural preservation. In addition, we demonstrate several applications of the protocol including the labeling of cell-type-specific proteins in various brain regions and proof-of-principle correlative light microscopy (LM)/EM experiments.

## Results

### Optimization of a permeabilization-free protocol to label ECS-preserved tissue sections

Aldehyde-based fixation of the brain has been previously demonstrated to result in the loss of the 15–20% ECS volume fraction normally found in vivo (*Van Harreveld and Malhotra, 1967*). The observation that detergent permeabilization is required for antibodies to penetrate into thick brain tissue sections has largely been based on such, usually perfusion-fixed, tissue in which ECS is not preserved. We began by demonstrating this requirement using a fluorophore-conjugated primary antibody for the neuronal soma marker, NeuN, as a representative antibody (*Mullen et al., 1992*) and prepared 300-μm-thick perfusion-fixed sections of mouse cerebral cortex. When detergent was omitted from the labeling protocol, we observed a gradient of fluorescently labeled somata that decreased in the center of sections, as expected, and the ultrastructural membrane integrity of the tissue remained intact (*Figure 1b*). With the inclusion of a commonly used detergent, Triton X-100, neuronal somata were uniformly labeled throughout the depth of the sections but at the cost of severely degraded membrane integrity due to lipid solubilization (*Figure 1a*).

We then replicated this experiment in 300-μm-thick acute immersion-fixed mouse cortical sections in which ECS was preserved as recently described (*Pallotto et al., 2015*). As with perfusion-fixed sections, the inclusion of a detergent in the protocol yielded uniform penetration of anti-NeuN at the expense of degraded membrane integrity (*Figure 1c*). However, when detergent was omitted from the protocol, the labeling of neuronal somata in ECS-preserved sections remained uniform throughout the tissue (*Figure 1d*).

Importantly, the exclusion of a permeabilization step yielded intact lipid membranes based on electron micrographs from the same tissue (*Figure 1d*). This result was repeatable in multiple cortical sections from two mice (*Figure 1—figure supplement 1*, see section 'Replicates'). We noted a nuclear exclusion of the NeuN labeling in the non-permeabilized sections (*Figure 1f*) compared to permeabilized sections (*Figure 1e*), indicating nuclear membranes remained impermeant to the antibody.

Based on the observation that anti-NeuN can penetrate thick ECS-preserved tissue sections without permeabilization, we proceeded to optimize the parameters of the protocol and explore the boundary conditions of the labeling uniformity. The parameters we titrated included fixative composition and duration, antibody concentration and duration of incubation, IHC buffer composition and osmolarity, incubation temperature, and the use of a tissue-clearing protocol. The resulting optimized IHC protocol (*Supplementary file 1*) incorporates these systematic optimization steps. A balance between ultrastructural integrity and penetration depth was dependent on the concentrations of paraformaldehyde (PFA) and glutaraldehyde (GA) in the primary fixative solution as expected based on the different cross-linking properties of the two aldehydes (*Hopwood, 1985*). A mixture of 4% PFA and 0.005% GA yielded a good compromise between penetration depth and EM quality, although small variations in the GA concentration led to similar results (*Figure 1—figure supplement 2*). A second key parameter was the concentration of the primary antibody. Uniform labeling was achieved when using IgG primary antibody concentrations in the range of 33–66 nM (*Figure 1—figure supplement 3a*). Of the parameters we varied, the labeling uniformity was less sensitive to the duration of the initial fixation (*Figure 1—figure supplement 3b*) and the omission of blocking serums (*Figure 1—figure supplement 4*), the utility of which has been questioned (*Buchwalow et al., 2011*). The ultrastructural quality was less sensitive to the duration of the initial

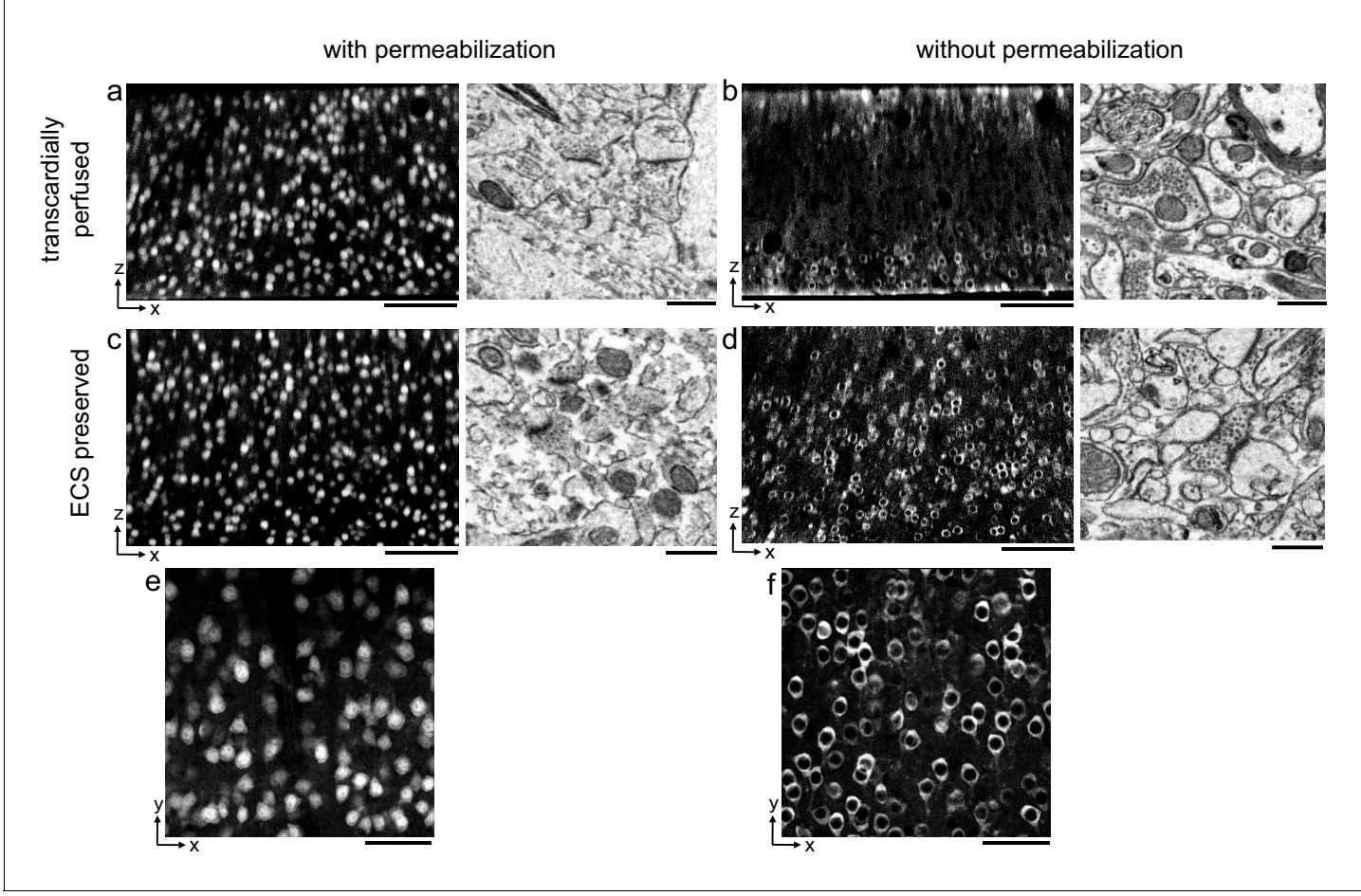

**Figure 1.** Permeabilization-free labeling of NeuN in extracellular space (ECS)-preserved cerebral cortex. (a, b) 300-µm-thick sections from the cerebral cortex of a transcardially perfused mouse. Incubation with Alexa Fluor-488 conjugated anti-NeuN was performed with (a) or without (b) 0.3% Triton to label neuronal somata. X–Z reslices of two-photon image volumes (left panels) are 10-µm-average intensity projections from the center of the sections. Sections were then stained for electron microscopy, and a region from the center of the section was examined for ultrastructural integrity (right panels). (c, d) Same procedure as in (a, b) but for 300-µm-thick acute ECS-preserved sections from the cerebral cortex of a mouse. (e, f) 10 µm average intensity projections of X–Y slices from center of the image volumes highlighting nuclear exclusion of anti-NeuN labeling when Triton is omitted (f). Scale bars: (a–d), left panels: 100 µm; (a–d), right panels: 1 µm; (e, f): 50 µm.

The online version of this article includes the following source data and figure supplement(s) for figure 1:

**Source data 1.** Table of tested primary and secondary antibodies.

**Figure supplement 1.** Replication of permeabilization-free labeling of NeuN across multiple tissue sections.

**Figure supplement 2.** Optimization of fixation parameters.

**Figure supplement 3.** Optimal antibody concentration and duration of primary fixation.

**Figure supplement 4.** Comparison of the effect of serum blocking on labeling thick sections.

**Figure supplement 5.** Comparison of the effect of prolonged antibody incubation on ultrastructure quality.

**Figure supplement 6.** Comparison of the effect of SeeDB clearing on ultrastructure quality.

fixation (*Figure 1—figure supplement 3b*) and the duration of the antibody incubations (*Figure 1—figure supplement 5*). Additional parameters were qualitatively determined: (1) the use of an antibody incubation buffer that was isotonic with the fixation buffer, (2) the ratio of NaCl to PB in the antibody incubation buffer, and (3) room temperature processing. For these three parameters, we observed adequate penetration and preservation of ultrastructure with the protocol described (*Supplementary file 1*), but we did not extensively explore these parameter spaces. It is possible that a titration of these three parameters may be more optimal for different antibodies or tissues.

Because we prepared relatively thick fixed tissue sections, we were unable to image fluorescence throughout the depth of the sections and therefore employed a tissue-clearing method. Many of the

tissue-clearing protocols recently described are not compatible with correlative LM/EM due to a degradation of tissue ultrastructure (*Richardson and Lichtman, 2015*). We therefore chose to use a refractive index-matching approach utilizing high-concentration fructose solutions, SeeDB, that does not involve the dissolution of membrane lipids (*Ke et al., 2013*). We confirmed that refixing the tissue with 2% PFA before serially incubating in fructose solutions retains ultrastructural quality (*Figure 1—figure supplement 6*). The use of SeeDB clearing allowed us to obtain two-photon (2P) excited fluorescence signals throughout the depth of 300 µm fixed tissue sections (*Denk et al., 1990*).

To explore the upper bound of the depth penetration achievable with the permeabilization-free protocol, we prepared ~1-mm-thick mouse cortical coronal sections and varied the fixative buffer concentration to yield different ECS volume fractions as previously described (*Pallotto et al., 2015*; *Figure 2*). The duration of the anti-NeuN incubation was increased from 72 hr (for 300 µm sections) to 120 hr to allow sufficient time for penetration. We observed that the depth of penetration increased as the fixative buffer concentration increased and that the highest concentration we used yielded complete penetration through 1-mm-thick sections (*Figure 2c*). As a result of the prolonged incubation duration, somatic nuclei near the edge of the section were occasionally labeled,

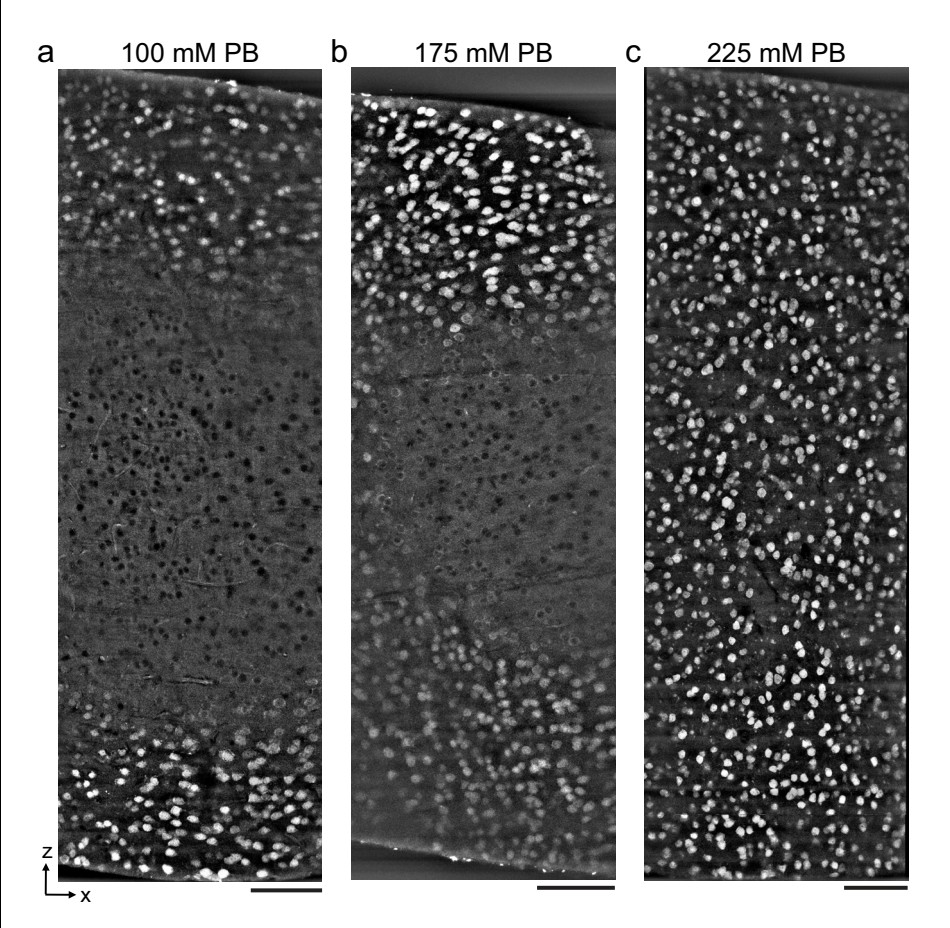

**Figure 2.** Antibody depth penetration increases with fixation osmolarity. Approximately 2 × 1 × 1-mm-thick acute extracellular space (ECS)-preserved sections from the mouse cerebral cortex, fixed with increasing buffer concentrations to yield (a) low (215 mOsm), (b) medium (360 mOsm), or (c) high (480 mOsm) osmolarities corresponding to increasing ECS volume fractions. Incubation with Alexa Fluor-488 conjugated anti-NeuN was performed without Triton to label neuronal somata. Tissue sections were hemisected and the cut surface was two-photon imaged to investigate the penetration depth. Images are 10 µm maximum intensity projections from the cut surface. Scale bars: 100 µm.

whereas those nuclei in the center exhibited exclusion of the antibody, indicating a time dependence of the nuclear exclusion.

### Distinguishing neuronal cell types and synaptic proteins with permeabilization-free IHC

During our optimization of the protocol parameters, we used anti-NeuN as a general label of neuronal somata. We next explored whether the protocol is compatible with additional antibodies and whether we could multiplex the immunofluorescent labeling of multiple protein targets. We first compared the labeling of neurons expressing the calcium-binding proteins calretinin (CR) and calbindin (CB) in the mouse cerebral cortex (*Figure 3a, b*). The CR+ and CB+ cell densities labeled with the permeabilization-free protocol (*Figure 3b*) were similar to those labeled when a permeabilization step was included (*Figure 3a*). Similar to the results with anti-NeuN, the permeabilization-free protocol generally yielded nuclear exclusion of the CR and CB labeling (*Figure 3b*).

We next examined the IHC labeling of the cytosolic enzyme choline acetyltransferase (ChAT) expressed in cholinergic neurons. We were motivated to explore whether cholinergic axons could be labeled in brain regions distant from their respective somata. Using tissue sections from the medial prefrontal cortex (mPFC), we achieved labeling of ChAT+ axons both with and without permeabilization (*Figure 3c, d*) consistent with previous reports of the innervation patterns in the mPFC (*Zhang et al., 2010*). However, we noted a discontinuity of the axonal labeling in the permeabilization-free sections (*Figure 3d*) compared to the permeabilized sections (*Figure 3c*). We hypothesize that the incomplete axonal labeling may be due to a narrowing of the diameter of portions of the axons and therefore present a hindrance to the diffusion of the antibodies or, potentially, that myelination may limit penetration into axons. Both of these possibilities are consistent with the lipid-

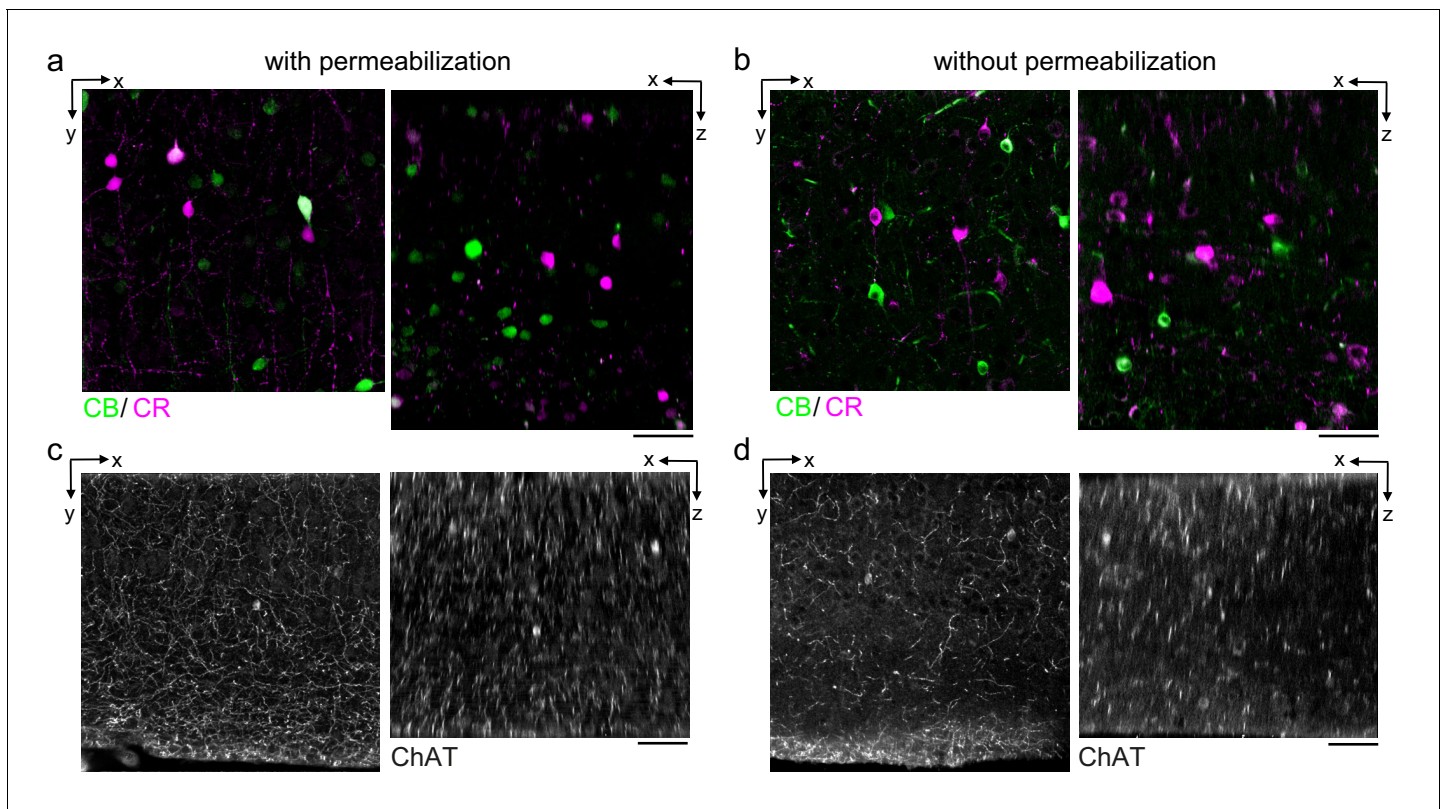

**Figure 3.** Permeabilization-free labeling of cell types and axons. (a, b) 300-μm-thick acute extracellular space (ECS)-preserved sections from the mouse cerebral cortex. Simultaneous incubation with anti-calbindin (CB) and anti-calretinin (CR) was performed with (a) and without (b) 0.3% Triton to label interneuron somata. X–Y slices (left panels) and X–Z reslices (right panels) of two-photon image volumes are 10 μm average intensity projections from the center of the sections. (c, d) Same as in (a, b) but 300-μm-thick acute ECS-preserved sections from the mouse medial prefrontal cortex were labeled with anti-choline acetyltransferase (ChAT) with (c) and without (d) 0.3% Triton to label cholinergic axons. Scale bars: 50 μm.

solubilizing effect of permeabilization that would improve penetration into axons, but at the cost of membrane integrity.

We then explored whether synaptic proteins in the mouse cortex could be labeled without permeabilization. We first attempted to label the postsynaptic density protein 95 (PSD-95), but found poor labeling without permeabilization (data not shown). Among postsynaptic proteins, super-resolution microscopy has demonstrated that PSD-95 is within approximately 30 nm of the postsynaptic membrane (*Dani et al., 2010*). We hypothesized that, in the absence of solubilizing lipids with Triton, there may be a steric hindrance for antibodies to label proteins that are in such close proximity to the plasma membrane. We therefore attempted to label Homer, a protein found more distant, greater than 50 nm, from the postsynaptic membrane of excitatory synapses (*Dani et al., 2010*). The labeling pattern of Homer (*Figure 4b*) was similar to permeabilized sections (*Figure 4a*) throughout the depth of 300-μm-thick sections. We noted a larger variability in the fluorescence intensity of labeled puncta in the non-permeabilized tissue compared to permeabilized tissue.

Finally, we explicitly compared the labeling efficiency of a presynaptic protein, synaptophysin, in the same tissue section before and after permeabilization. We took advantage of the availability of synaptophysin antibodies directly conjugated to Alexa Fluor (AF) molecules of different excitation wavelengths (AF488 and AF594). We first labeled mouse cortical sections with AF488 anti-synaptophysin without permeabilization and then labeled the same sections with AF594 anti-synaptophysin including Triton permeabilization. Prior to permeabilization, we observed a labeling density consistent with published labeling patterns (*Figure 4c*, green) (*Grant et al., 2016*). Following permeabilization, the labeling pattern was similar at the edges of the tissue but the antibody labeling was notably absent in the center of the section (*Figure 4c*, magenta). This result is consistent with the observation that treatment with Triton X-100 can lead to the extraction of synaptophysin from fixed tissue (*Hannah et al., 1998*). To distinguish whether any presynaptic puncta were labeled following permeabilization that were not labeled prior to permeabilization, we quantified the co-localization of puncta in the two fluorescence channels (*Figure 4d*). Puncta intensities (n = 150 puncta) were significantly correlated (linear regression t-test, p=$1.5\times10^{-8}$) compared to a 90° rotation of the images (p=0.86), which indicated the co-localized labeling of synaptophysin pre- and post-permeabilization.

## Applicability of permeabilization-free labeling across different brain regions

In addition to the mouse cerebral cortex, we assessed antibody penetration in brain regions with different cytoarchitectures including the mouse hippocampus and olfactory bulb. Labeling 300-μm-thick tissue sections from these brain regions with antibodies targeting NeuN, CR, and CB resulted in patterns similar to permeabilized sections (*Figure 5*). In the hippocampus, we observed the CR + subgranular band and granular cell layer positioning of CB+ neurons as previously reported (*Ohira et al., 2010*; *Figure 5a–d*). In the olfactory bulb, non-overlapping populations of CB+ and CR + periglomerular cells (*Panzanelli et al., 2007*) were positively identified (*Figure 5e–h*).

## Permeabilization-free labeling for correlative LM/EM

During the optimization of the protocol, we utilized small organic dyes conjugated to primary or secondary antibodies as reporters of positive labeling. We also tested whether a larger enzyme, such as horseradish peroxidase (HRP), would be compatible with permeabilization-free IHC to enable the direct correlation between fluorescence and an electron dense product. We therefore investigated the conversion of eGFP expressed in neurons into an electron dense signal using HRP conjugated to a secondary antibody. We retrogradely labeled mitral and tufted cells in the mouse olfactory bulb by injecting an adeno-associated virus (AAV) encoding Flex-eGFP in the mouse piriform cortex of a Cdhr1-Cre mouse (*Wachowiak et al., 2013*). Following 2P imaging of 300-μm-thick acute sections (*Figure 5i*), the eGFP was labeled with a primary anti-GFP antibody and a secondary antibody conjugated to HRP. The HRP-assisted polymerization of diaminobenzidine (DAB) was then carried out in the presence of hydrogen peroxide, yielding a DAB polymer that was strongly osmiophilic in EM images. Individual mitral and tufted cell neurites were readily distinguished from unlabeled neurites in the surrounding neuropil of the external plexiform layer of the olfactory bulb (*Figure 5i*).

Our primary motivation for developing a permeabilization-free IHC protocol was to enable the correlation of fluorescently labeled endogenous proteins with their location in large-volume EM

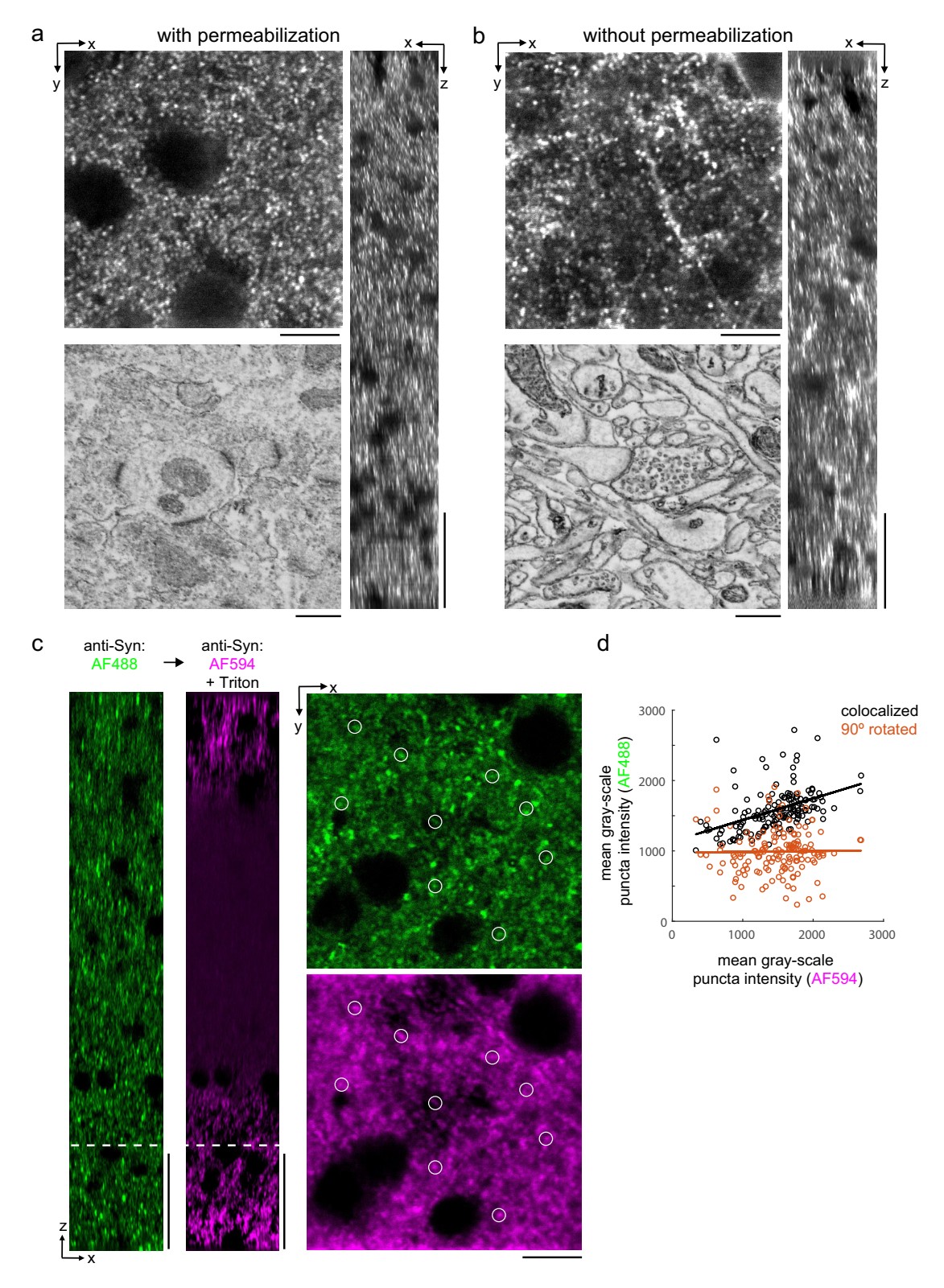

**Figure 4.** Permeabilization-free labeling of synaptic proteins. (**a, b**) 300-µm-thick acute extracellular space (ECS)-preserved sections from the mouse cerebral cortex. Incubation with anti-Homer was performed with (**a**) and without (**b**) 0.3% Triton. X–Y slices (upper-left panels) and X–Z reslices (right panels) of two-photon image volumes are taken from the center of the sections. Sections were then stained for electron microscopy, and a region from the center of the section was examined for ultrastructural integrity (lower-left panels). (**c**) 300-µm-thick acute ECS-preserved slices from the mouse

*Figure 4 continued on next page*

*Figure 4 continued*

cerebral cortex. Sequential incubation with Alexa Fluor-488 (AF488) conjugated anti-synaptophysin (anti-Syn) without Triton (left, green) followed by Alexa Fluor-594 (AF594) conjugated anti-Syn with 0.3% Triton (left, magenta) shown as X–Z reslices. Puncta were outlined in X–Y images at the depth indicated (left, white dashed line) in the AF594 channel (lower-right panel, representative white circles) for co-localization analysis with the AF488 channel (upper-right panel). (**d**) Co-localization analysis of 150 puncta from image stacks in (**c**) comparing co-localized puncta (black) to a 90° rotation of the AF488 channel (orange). Scale bars: (**a, b**) (upper-left panels: 10 μm; right panels: 50 μm; lower-left panels: 0.5 μm), (**c**) (left panels: 50 μm; right panels: 10 μm).

data. As a first proof of principle, we investigated the application of identifying fluorescently labeled somata deep within a tissue section in a corresponding EM ultrathin section. We performed triple immunolabeling of CR+, CB+, and tyrosine hydroxylase (TH+) interneurons in a 300 μm section from the mouse olfactory bulb (*Figure 6a*, left panel). We then processed the same section for EM and cut ultrathin slices until reaching a depth of approximately 150 μm in the section (*Figure 6b*, left panel). We collected an ultrathin section at this depth and, using the fluorescently labeled somas as landmarks, matched soma locations between the 2P image stack and the EM slice. An affine transform was applied to the 2P image stack to register it to the EM slice, and we were then able to identify somas in the EM slice that corresponded to the immunolabeled somata in a slice from the transformed 2P stack (*Figure 6a, b*, right panels).

Ultimately, the correlation of LM and EM is most informative when performed in 3D volumes. We therefore combined the permeabilization-free IHC protocol with a block-face EM technique as a correlative pipeline consisting of five steps (*Supplementary file 1*): (1) permeabilization-free *en bloc* IHC, (2) tissue clearing by refractive index matching (*Ke et al., 2013*), (3) 2P imaging of the tissue volume and near-infrared branding (*Bishop et al., 2011*) of a region of interest, (4) reversal of the tissue-clearing protocol back to buffer, and (5) EM staining and acquisition of a SBEM volume.

Using this pipeline, as a second proof of principle, we labeled axons endogenously expressing TH in a 300 μm section from the mouse mPFC (*Zhang et al., 2010*; *Figure 6c*). Following SeeDB clearing, 2P imaging and branding of the section, we stained the tissue for EM and collected an 89 × 83 × 75 μm³ SBEM volume centered on the branded region that began approximately 50 μm deep into the section (*Figure 6c*). The 2P and EM datasets were aligned by fitting an affine transform using landmarks in the two datasets including somata and blood vessels, similar to an approach recently developed to match axons expressing fluorescent proteins (*Drawitsch et al., 2018*). We then locally searched regions of the EM dataset to identify the matching trajectories of fluorescent axons to those in the EM volume (*Figure 6d, e*) and identified TH+ axons within the EM volume (*Figure 6f, g*). The traced axons spanned a depth of 57–70 μm deep within the tissue section.

## Discussion

We have developed a protocol to immunohistochemically label thick tissue sections that omits the commonly used permeabilization step and is therefore compatible with correlative *en bloc* volume EM techniques. The method depends on the preservation of ECS during tissue fixation (*Figures 1* and *2*), and we suggest that the simplest explanation for the improved labeling is that the diffusion spaces that are preserved allow antibodies to diffuse deep into tissue and travel to close proximity of their antigens. That is, rather than translocating across multiple plasma membranes in densely packed neuropil, an antibody in ECS-preserved tissue could, in principle, only need to cross one membrane to reach an epitope. The mechanism by which an antibody crosses a membrane that has not been permeabilized by a detergent is not clear to us, but we note a few observations. First, aldehyde fixation alone has been described to semi-permeabilize membranes due to their denaturing effects on proteins (*Hopwood, 1985*). Second, the hydrodynamic radius of an IgG antibody is approximately 5 nm (*Armstrong et al., 2004*; *Jøssang et al., 1988*), meaning a small pore in a membrane of similar size is required to provide access to intracellular epitopes. We cannot rule out that the ECS preservation process itself leads to damage of some membranes allowing an antibody to enter through a damaged region and then diffuse along the intracellular cytosol of neurons. However, if such damage occurs, it has not prevented us from reconstructing neuronal morphologies and identifying synapses in ECS-preserved tissue (*Pallotto et al., 2015*; *Figures 4* and *6*). Applying the method to high-pressure frozen tissue in which ultrastructure and ECS is preserved in a more native

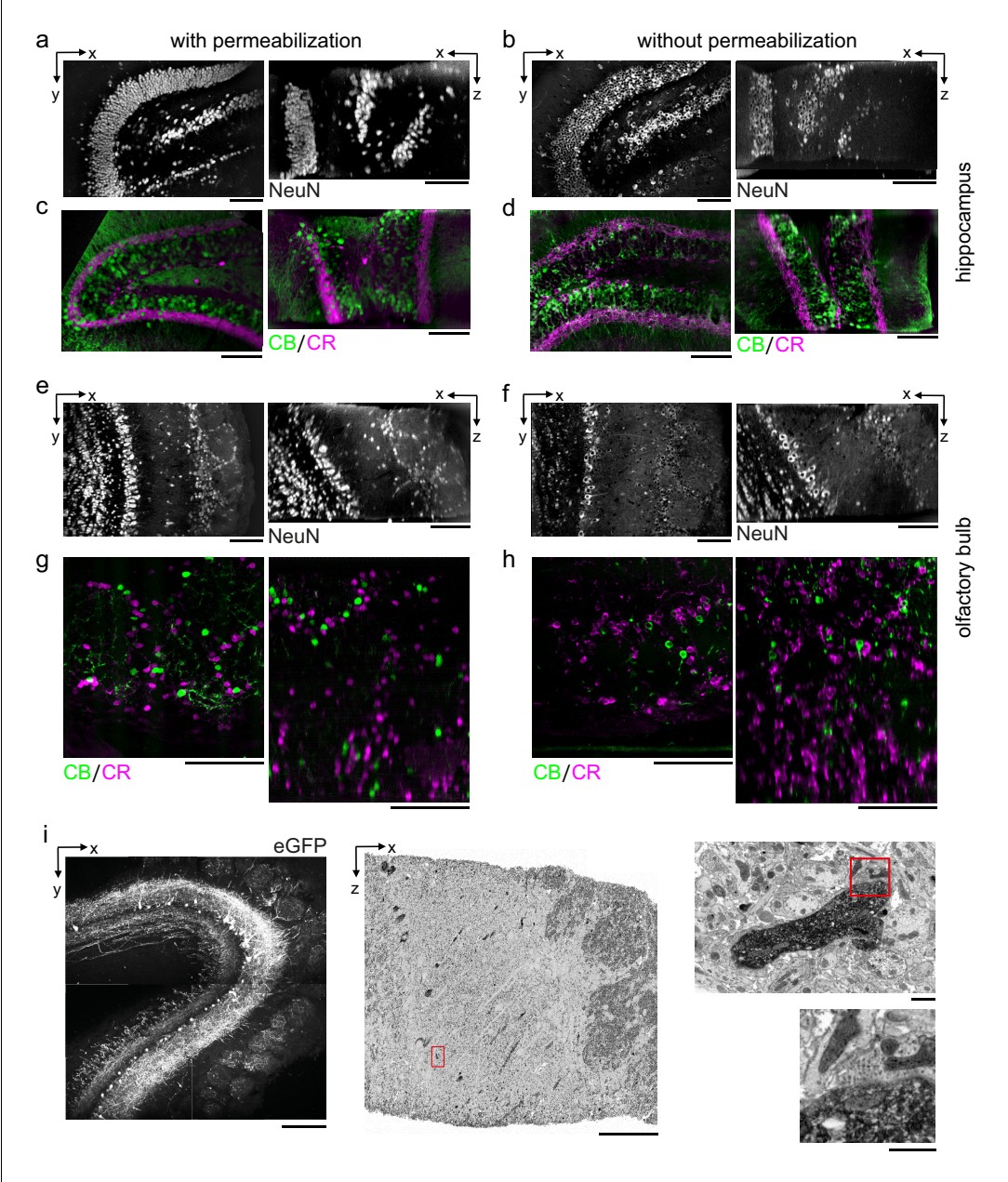

**Figure 5.** Permeabilization-free labeling of diverse brain regions and genetically expressed proteins. (**a–d**) 300-μm-thick acute extracellular space (ECS)-preserved sections from the mouse hippocampus. Incubation with Alexa Fluor-488 conjugated anti-NeuN (**a, b**) or simultaneous incubation with anti-calbindin (CB) and anti-calretinin (CR) (**c, d**) was performed with (**a, c**) and without (**b, d**) 0.3% Triton to label interneuron somata. X–Y slices (left panels) and X–Z reslices (right panels) of two-photon (2P) image volumes are 10 μm average intensity projections from the center of the sections. (**e–h**) 300-μm-thick acute ECS-preserved sections from the mouse olfactory bulb. Incubation with Alexa Fluor-488 conjugated anti-NeuN (**e, f**) or simultaneous incubation with anti-CB and anti-CR (**g, h**) was performed with (**e, g**) and without (**f, h**) 0.3% Triton to label neuronal somata. 2P image slices as in (**a–d**). (**i**) Horseradish peroxidase (HRP) labeling of eGFP expressing mitral cells (MCs) in the mouse olfactory bulb. 300-μm-thick acute ECS-preserved sections containing MCs expressing eGFP following adeno-associated virus transfection (left panel). Incubation with anti-GFP and HRP-conjugated secondary antibody was performed without Triton, followed by polymerization of diaminobenzidine (DAB) and electron microscopy staining (middle panel). Enlarged regions (red rectangles) demonstrate confinement of the DAB product to MC dendrites (right, upper panel) and a presynaptic terminal formed on a labeled dendrite (right, lower panel). Scale bars: (**a–h**) 100 μm; (**i**) 200 μm (left), 50 μm (middle), 1 μm (right).

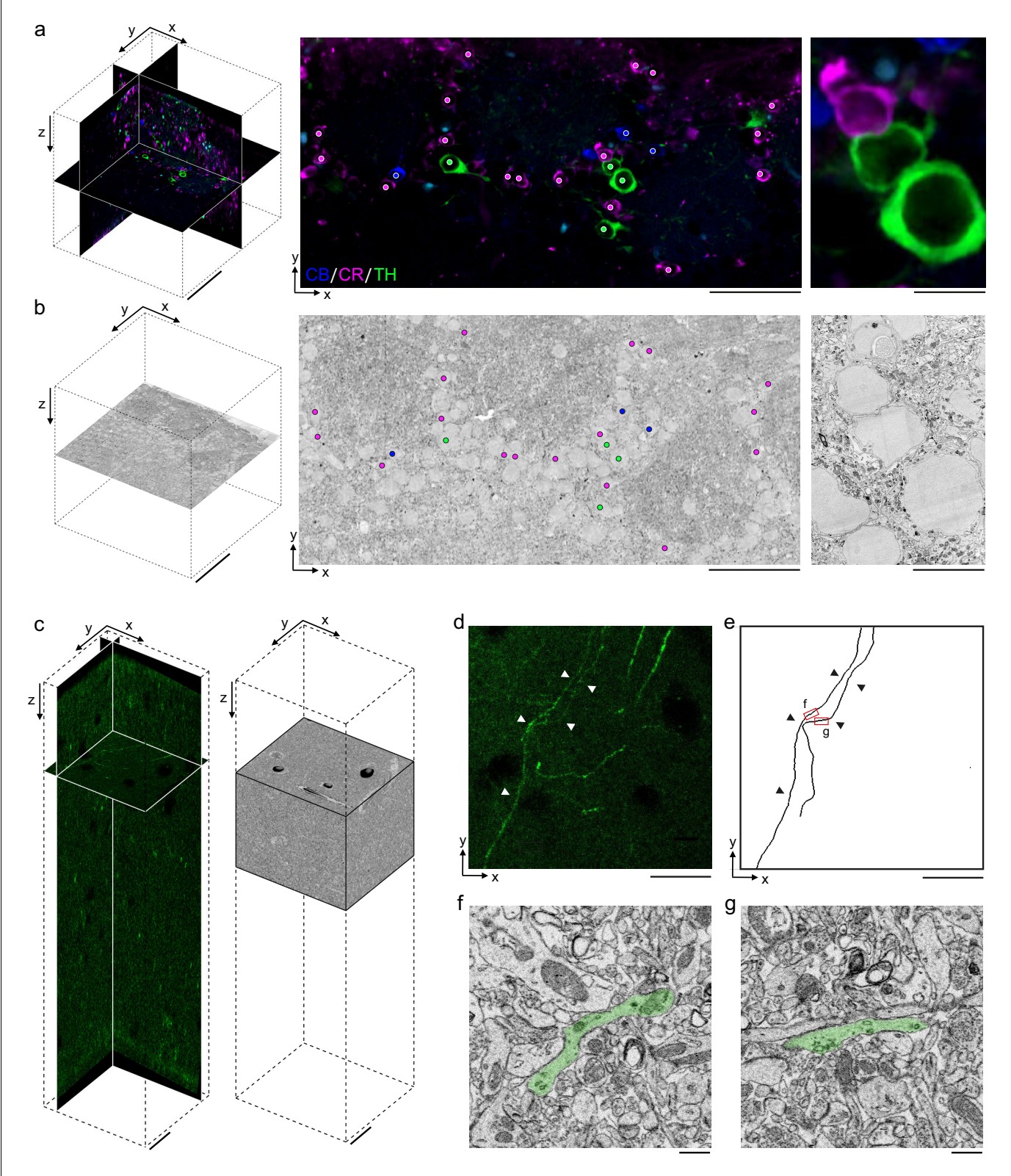

**Figure 6.** Permeabilization-free labeling for correlative microscopy. (**a, b**) Correlative microscopy of calbindin-positive (CB+), calretinin-positive (CR+), and tyrosine hydroxylase-positive (TH+) interneurons in the mouse olfactory bulb. (**a**) 300-μm-thick acute extracellular space (ECS)-preserved section (left panel) incubated with anti-CB (blue), anti-CR (magenta), and anti-TH (green) without Triton. An X–Y slice (middle panel) from the two-photon (2P) image volume located ~150 μm deep in the section. Colored dots indicate the positive labeling by the respective antibodies. Enlarged region

*Figure 6 continued on next page*

*Figure 6 continued*

illustrates three somata (right panel). (**b**) 35 nm electron microscopy (EM) slice cut from ~150 μm deep in the section from panel (**a**) (left panel). Colored dots indicate correspondence between soma locations in the EM section and the 2P section (middle panel). Enlarged region illustrates three somata (right panel). (**c–g**) Correlative microscopy of TH+ axons in the mouse medial prefrontal cortex. (**c**) 2P image stack from a 300-μm-thick ECS-preserved acute section that was incubated with anti-TH without Triton to label dopaminergic axons (left panel). A serial block-face scanning electron microscopy (SBEM) volume centered on branded fiducial marks spans a depth of 50–125 μm in the 2P image stack (right panel). (**d**) A 10-μm-thick X–Y maximum intensity projection from panel (**c**) with TH+ axons indicated (white arrowheads). (**e**) Anatomical reconstructions of corresponding TH+ axons from the SBEM volume with matching axons indicated (black arrowheads). (**f, g**) Example EM images of the reconstructed axons from the regions indicated in panel (**e**). Axons are false-colored (green). Scale bars: (**a, b**) 100 μm (left), 50 μm (middle), 10 μm (right); (**c–e**) 20 μm; (**f, g**) 1 μm.

state may yield additional insights into the mechanism (*Korogod et al., 2015*). Freeze-thaw cycles (cryo-permeabilization) have been used to permeabilize single cells for IHC, but we are not aware of such a protocol being used to permeabilize thick tissue sections while still maintaining acceptable ultrastructural preservation.

A current limitation of the protocol is our inability to label antigens that are closely associated with lipid membranes such as the postsynaptic protein PSD-95. The postsynaptic terminal is densely packed with protein, and our ability to label the Homer protein, which is more distant from the post-synaptic membrane than PSD-95 (*Dani et al., 2010*), suggests a steric hindrance for antibody binding when a protein is in close proximity to the plasma membrane. In contrast, we were able to label the presynaptic protein synaptophysin located on the surfaces of vesicles that are relatively accessible within presynaptic terminals. Therefore, some degree of permeabilization may be a requirement for proteins that are located in close spatial proximity to the plasma membrane. One alternative would be to explore weaker detergents than Triton, such as low concentrations of Tween, which was previously used to label ECS-preserved retinal tissue with satisfactory ultrastructure (*Pallotto et al., 2015*). Alternatively, if the size of IgG antibodies is limiting the binding to postsynaptic targets, the use of physically smaller antibodies, such as nanobodies (*Hamers-Casterman et al., 1993*), may improve labeling in combination with our protocol. Nanobodies have recently been shown to label targets up to 100 μm, but not deeper, from one surface of a tissue section (*Fang et al., 2018*). The use of nanobodies in the future may also further improve the labeling of axonal proteins (*Figure 3*) and perhaps transcription factors. Alternatively, organic dyes that bind to fusion proteins have been shown to facilitate the super-resolution localization of proteins combined with FIB-SEM (*Hoffman et al., 2020*). The diffusion of such small molecules would likely allow penetration into thick tissue sections as well, but, because such an approach requires the expression of a fusion protein, it is not compatible with the labeling of endogenous proteins that we have focused on here. A further limitation of our protocol is the reliance on immersion fixed tissue sections for ECS preservation. While we focused on section thicknesses up to 1 mm, an important future goal is to achieve ECS preservation by perfusion of a whole brain (*Cragg, 1980*) and uniform permeabilization-free antibody labeling throughout large tissue volumes.

To replicate the protocol, we emphasize that it is important to first confirm that ECS preservation during acute section immersion fixation was successful, ideally by inspection of EM images. The degree of ECS preservation varies by brain region as previously reported (*Pallotto et al., 2015*) and needs to be calibrated for specific brain regions. The protocol also requires the typical optimization of IHC conditions for a given antibody such as concentration, buffer composition, and duration (*Supplementary file 1*). We have focused on proteins with known expression patterns for the optimization of the protocol. Our strategy was to compare the labeling pattern in permeabilized tissue to that of ECS preserved non-permeabilized tissue. For novel proteins, labeling specificity should be assayed as previously described (*Lorincz and Nusser, 2008*). We have not yet assessed whether antigen retrieval steps with proteases such as pepsin would further improve labeling while still maintaining ultrastructure (*Watanabe et al., 1998*) or whether multiplexing by repeated elution and relabeling, such as in array tomography (*Micheva and Smith, 2007*), is compatible with the protocol.

The ultimate test of the utility of this protocol is in 3D correlative LM/EM reconstructions, for which we demonstrated two proof-of-principle applications (*Figure 6*). We anticipate that permeabilization-free pre-embedding IHC could play an important role in adding functional information to anatomical datasets. The combination of cellular-resolution functional imaging, permeabilization-free

IHC, and *en bloc*-based EM all within the same tissue would enable the correlation of function, cell-type identity, neuronal morphology, and synaptic connectivity within local circuits. Our protocol could also be further optimized for tissue from species in which fluorescent genetic tagging of proteins is not possible, including human tissue obtained from immersion fixed biopsies.

## Materials and methods

### Tissue fixation

C57BL/6J male mice (age p30–p45) were used for all experiments except for the use of *Cdhr1-cre* mice (Gensat stock # 030952-UCD; *Nagai et al., 2005*), age p50, to label mitral and tufted cells (*Figure 5*) and *Olfr160-IRES-ChR2-YFP* mice (JAX stock # 021206; *Smear et al., 2013*), age p42, to label interneurons in the mouse olfactory bulb (*Figure 6*). All animal procedures were conducted in accordance with US National Institutes of Health guidelines, as approved by the National Institute of Neurological Disorders and Stroke Animal Care and Use Committee (ASP 1340).

For transcardially perfused tissue fixation, we followed a rapid perfusion approach (*Tao-Cheng et al., 2007*). Briefly, mice were anesthetized with isofluorane (Forane) and then perfused with 4% PFA (Electron Microscopy Services) and 0.005% GA (Electron Microscopy Sciences) in 175 mM sodium phosphate buffer (PB, pH 7.4). The brain was extracted and post-fixed for 4 hr in the same fixative solution. The perfused brain was subsequently rinsed in 175 mM PB for 4 hr and 300 μm coronal sections were cut on a Leica Vibratome.

For acute ECS-preserved sections, mice were first anesthetized with isofluorane before swift decapitation. The brain was carefully removed from the skull, and 300 μm (up to 1 mm) coronal sections from the olfactory bulb, cerebral cortex, and/or hippocampus were cut on a vibratome (Leica) according to the procedure of *Bischofberger et al., 2006* and briefly stored in a cold carboxygenated (95% $O_2$/5% $CO_2$) ACSF solution (300–320 mOsm) containing (in mM): 124 NaCl, 3 KCl, 1.3 $MgSO_4.7H_2O$, 26 $NaHCO_3$, 1.25 $NaH_2PO_4.H_2O$, 20 glucose, 2 $CaCl_2.2H_2O$. Sections were then immersion-fixed using a protocol to preserve the ECS (*Pallotto et al., 2015*). The fixative concentrations were varied (see *Figure 1—figure supplement 2*), and an optimal primary fixative of 4% PFA + 0.005% GA in 175 mM PB (pH 7.4, 4°C, 360 mOsm) for 4 hr was chosen. The total duration between decapitation and fixation was less than 10 min.

### Immunohistochemistry

Primary and secondary antibody catalog numbers and incubation concentrations are listed in *Figure 1—source data 1*. Optimal antibody concentrations were titrated for each antibody tested. For antibodies with stock concentrations of 0.5 mg/mL, antibody dilutions were typically 1:100 or 1:50. Antibody concentrations in nM were calculated using an IgG molecular weight of 150 kg/mol, the stock concentration and the dilution factor. All antibodies were diluted in an antibody buffer solution (ABS) of 170 mM NaCl + 10 mM PB (360 mOsm). 0.3% Triton X-100 (Electron Microscopy Sciences) was added to both the primary and secondary antibody solution for sections that were permeabilized. Sections were added to individual wells of a 96-well plate, each containing 50–75 μL ABS. Care was taken to ensure that the sections were fully immersed in the ABS and not adhered to the walls prior to wrapping the plate in Parafilm. All IHC steps were carried out in the dark at room temperature on an orbital shaker at medium-high speed. The duration of primary and secondary antibody incubation was dependent on section thickness. Typically 300 μm sections were incubated in primary antibody for 72 hr and in secondary antibody for 48 hr.

### Somatic labeling (300-μm-thick sections)

Following fixation, sections were rinsed in 175 mM PB for 4 hr, then a buffered glycine solution (50 mM glycine in 175 mM PB) for 16 hr, and then rinsed for 6 hr in 175 mM PB. Regions of 300-μm-thick coronal sections were then trimmed down to volumes of approximately 2 × 2 × 0.3 mm³ with a scalpel for cortical and hippocampal regions; olfactory bulb sections were left intact. For labeling of NeuN-positive somata (*Figures 1* and *5a, b, e, f*), sections were incubated in the ABS with Alexa Fluor-488 conjugated anti-NeuN (Abcam) for 72 hr. For labeling of CB+ and CR+ somata (*Figures 3a–b* and *5c, d, g, h*), sections were simultaneously incubated in guinea pig anti-CB (Synaptic Systems) and rabbit anti-CR (Sigma) for 72 hr, rinsed in 175 mM PB for 12 hr, and then incubated

in the secondary antibodies, DyLight-405 goat anti-guinea pig IgG F(ab')$_2$ fragment (Jackson ImmunoResearch) for CB and Alexa Fluor-594 donkey anti-rabbit IgG F(ab')$_2$ fragment (Jackson ImmunoResearch) for CR, for 48 hr.

## Somatic labeling (1-mm-thick sections)
Coronal sections were fixed in 4% PFA + 0.005% GA in either 100 mM, 175 mM, or 225 mM PB (pH 7.4, 4°C) for 4 hr. Following fixation, sections were rinsed for 4 hr in PB of the respective concentration used for fixation, then a buffered glycine solution (50 mM glycine also in the respective PB concentration) for 16 hr, and then rinsed for 6 hr in PB of the respective concentration. 1-mm-thick coronal sections were trimmed down to volumes of approximately $4 \times 1 \times 1$ mm$^3$ with a scalpel and incubated in Alexa Fluor-488 conjugated anti-NeuN (Abcam) for 120 hr (*Figure 2*). Sections were then embedded in 10% quick dissolve agarose (GeneMate) and hemisected to a volume of approximately $2 \times 1 \times 1$ mm$^3$ to observe anti-NeuN penetration through the 1 mm depth with 2P microscopy on the hemisected surface.

## Axon labeling
Following fixation, sections were rinsed in 175 mM PB for 4 hr, then a buffered glycine solution (50 mM glycine in 175 mM PB) for 16 hr, and then rinsed for 6 hr in 175 mM PB. Regions of 300-μm-thick coronal sections were then trimmed down to volumes of approximately $2 \times 2 \times 0.3$ mm$^3$ with a scalpel. For labeling of ChAT-positive axons (*Figure 3c–d*), sections were incubated in goat anti-ChAT (EMD Millipore) for 72 hr, rinsed in 175 mM PB for 8 hr, and then incubated in DyLight-594 donkey anti-goat (Abcam) for 48 hr. For labeling of TH+ axons (*Figure 6c*), sections were incubated in Alexa Fluor-488 conjugated anti-TH, clone LNC1 (EMD Millipore) for 72 hr, and then rinsed in 175 mM PB for 8 hr.

## Synaptic protein labeling
Following fixation, sections were rinsed in 175 mM PB for 4 hr, then a buffered glycine solution (50 mM glycine in 175 mM PB) for 16 hr, and then rinsed for 6 hr in 175 mM PB. Regions of 300-μm-thick coronal sections were then trimmed down to volumes of approximately $2 \times 2 \times 0.3$ mm$^3$ with a scalpel. For labeling of Homer in postsynaptic terminals (*Figure 4a, b*), sections were incubated in rabbit anti-Homer 1 (Synaptic Systems) for 72 hr, rinsed in 175 mM PB for 8 hr, and then incubated in Alexa Fluor-594 donkey anti-rabbit IgG F(ab')$_2$ fragment (Jackson ImmunoResearch) for 48 hr. For labeling of synaptophysin in presynaptic terminals (*Figure 4c*), sections were first incubated in Alexa Fluor-488 conjugated rabbit anti-synaptophysin (Abcam) for 72 hr, rinsed in 175 mM PB for 24 hr, and then incubated in Alexa Fluor-594 conjugated rabbit anti-synaptophysin (Abcam) plus 0.3% Triton for 48 hr.

## Serum blocking
For sections in which serum blocking was employed (*Figure 1—figure supplement 4*), tissue was first incubated in a blocking buffer of 10% normal donkey serum, 1% bovine serum albumin, and 0.05% sodium azide in 175 mM PB for 4 hr. Following the serum blocking step, sections were incubated with antibodies as described above with 3% normal donkey serum.

## HRP labeling of eGFP
Cre-dependent expression of eGFP in axons and dendrites of mitral cells was achieved by injection of AAV encoding eGFP into the anterior piriform cortex. AAV pCAG-FLEX-EGFP-WPRE was a gift from Hongkui Zeng (Addgene plasmid # 51502-AAV1; http://n2t.net/addgene:51502; RRID:Addgene_51502) (*Oh et al., 2014*). Under isofluorane anesthesia, age p30 Cdhr1-Cre mice were positioned into a stereotaxic head holder, and a small craniotomy was made on the dorsal surface above the injection site. A glass pipette was used to inject the virus, targeting axon terminals of mitral cells bilaterally in the anterior piriform cortex (350 nL per side). After 3 weeks, animals were sacrificed and 300-μm-thick acute ECS-preserved coronal sections of the olfactory bulb were prepared as above. Following fixation, sections were rinsed in 175 mM PB for 4 hr, then a buffered glycine solution (50 mM glycine in 175 mM PB) for 16 hr, and finally rinsed for 6 hr in 175 mM PB. eGFP fluorescence in the coronal sections was first imaged with a 2P microscope without SeeDB clearing (see

below). Subsequently, cytosolic eGFP in mitral and tufted cells was labeled for EM (*Figure 5i*) by incubating sections in rabbit anti-GFP (ThermoFisher) for 72 hr, rinsing in 175 mM PB for 8 hr, and then incubating in HRP-conjugated donkey anti-rabbit (Abcam) for 48 hr. Sections were refixed with 2% GA in 150 mM PB for 2 hr at room temperature, rinsed in 150 mM PB for 2 hr, rinsed in glycine (50 mM glycine in 175 mM PB) for 16 hr, rinsed again in 150 mM PB for 2 hr, and then incubated at room temperature in the dark on a rotator in 1.4 mM diaminobenzidine hydrotetrachloride (DAB, Serva) and 0.56 mM hydrogen peroxide (Sigma) in 175 mM PB for 10 hr to polymerize DAB in the presence of HRP.

## Triple labeling of olfactory bulb

Following fixation, sections were rinsed in 175 mM PB for 4 hr, then a buffered glycine solution (50 mM glycine in 175 mM PB) for 16 hr, and then rinsed for 6 hr in 175 mM PB. For labeling of CB+, CR +, and TH+ somata (*Figure 6a*), sections were first simultaneously incubated with guinea pig anti-CB (Synaptic Systems) and rabbit anti-CR (Sigma) for 72 hr, then rinsed in 175 mM PB for 72 hr. Sections were then simultaneously incubated with DyLight-405 goat anti-guinea pig IgG F(ab')$_2$ fragment (Jackson ImmunoResearch) for CB, Alexa Fluor-594 donkey anti-rabbit IgG F(ab')$_2$ fragment (Jackson ImmunoResearch) for CR, and the primary antibody Alexa Fluor-488 conjugated anti-TH, clone LNC1 (EMD Millipore) for 72 hr.

## Tissue clearing

Following antibody labeling, sections were rinsed in 175 mM PB for 16 hr, refixed in buffered PFA (2% PFA in 175 mM PB) for 2 hr, and rinsed in 175 mM PB for 4 hr. Sections were subsequently incubated in a modified SeeDBp protocol (*Ke et al., 2013*): 20% w/v fructose in 175 mM PB for 2 hr, 40% w/v fructose in 175 mM PB for 3 hr, 60% w/v fructose in 175 mM PB for 4 hr, 80% w/v fructose in 175 mM PB for 4–6 hr, 100% w/v fructose in water for a minimum of 10 hr, and finally in SeeDB solution (80.2% w/w fructose in water and 0.5% 1-thioglycerol) for a minimum of 10 hr. Fructose incubation was omitted for 1-mm-thick sections because the hemisected surface was imaged directly to assess antibody penetration. This was done to rule out that a potential labeling gradient would be due to light scattering rather than limited antibody penetration. With custom objectives matched to the refractive index of the clearing solution, SeeDB has been successfully utilized for 2–6-mm-thick sections (*Ke et al., 2013*).

## Two-photon imaging

Immunolabeled sections were mounted in SeeDB solution between two #1 cover slips in a custom imaging chamber and imaged with a ×20, 1.0 NA water immersion objective (Olympus) on a 2P laser scanning microscope (Sutter) using ScanImage (*Pologruto et al., 2003*). An excitation wavelength of 770–800 nm was used to image sections incubated with Alexa Fluor-488 and Alexa Fluor-594/ DyLight-594. An excitation wavelength of 770 nm was used to image the DyLight-405 fluorophore. 2P image stacks were collected with 0.5–1 μm z-step sizes throughout the depth of 300-μm-thick sections at various x–y pixel sizes.

## Near-infrared branding

A near-infrared branding technique (*Bishop et al., 2011*) was used to burn fiducial marks 20–30 μm below the top surface of sections around a region of interest (*Figure 6c*) after 2P image stack acquisition, typically a 150 × 150 μm$^2$ region. Following branding, sections were gradually returned to 175 mM PB in decreasing fructose concentrations over 16 hr and then post-fixed in 2% GA in 150 mM sodium cacodylate buffer (CB, Electron Microscopy Sciences) for 2 hr.

## Tissue processing for EM

For sections prepared for correlative EM, sections were thoroughly rinsed in 175 mM CB prior to EM staining. Sections were stained using a modification of previously described protocols (*Briggman et al., 2011*; *Karnovsky, 1971*). Briefly, tissue was stained in a solution containing 2% osmium tetroxide, 3% potassium ferrocyanide, and 2 mM CaCl$_2$ in 150 mM CB for 2 hr at 4°C. The osmium stain was amplified with 1% thiocarbohydrazide (1 hr at 50°C) and 2% osmium tetroxide (1 hr at room temperature). The tissue was then stained with 1% aqueous uranyl acetate for 6 hr at 45°

C and lead aspartate for 6 hr at 45℃. The tissue was dehydrated at 4℃ through an ethanol series (70%, 90%, 100%), transferred to propylene oxide, infiltrated at room temperature with 50%/50% propylene oxide/Epon Hard, and then 100% Epon Hard (*Glauert and Lewis, 1998*). The blocks were cured at 60℃ for 24 hr. Ultrathin sections (35–70 nm) were cut from block faces and imaged on a NanoSEM 450 (FEI) or multibeam SEM (Zeiss).

## Serial block-face electron microscopy

For SBEM acquisition (*Figure 6c*), data were collected using a custom serial block-face micro-tome designed by K.L. Briggman. The specimens were cut out of the flat-embedding blocks and re-embedded in Epon Hard on aluminum stubs. The samples were then trimmed to a block face of ~200 μm wide and ~200 μm long. The samples were imaged in a scanning electron microscope with a field-emission cathode (NanoSEM 450, FEI). Back-scattered electrons were detected using a concentric segmented back-scatter detector. The incident electron beam had an energy of 2.4 keV and a current of ~200 pA. Images were acquired with a pixel dwell time of 2 μs and size of 10.25 nm × 10.25 nm. Imaging was performed at high vacuum, with the sides of the block sputter coated with a 100-nm-thick layer of gold. The section thickness was set to 32 nm. 2354 consecutive block faces were imaged from the sample, resulting in aligned data volume of 8704 × 8064 × 2354 voxels (corresponding to an approximate spatial volume of 89 × 83 × 75 $μm^3$). Axons in the SBEM volume corresponding to TH+ axons in the 2P stack were annotated using Knossos (https://knossos.app/) (*Helmstaedter et al., 2011*).

## Replicates

Biological replicates for *Figure 1* are shown in *Figure 1—figure supplement 1*. 10 cortical sections from one mouse were perfusion-fixed and 10 sections from a second mouse were acutely fixed with ECS preservation. Of the 10 sections from each mouse, 5 were treated with Triton and 5 were not permeabilized during IHC incubation. For *Figures 2*, *3*, and *5, a* minimum of three mice, each with a minimum of three sections for each condition, were performed. Synaptic labeling (*Figure 4a, b*) was performed in six cortical sections from one mouse. Synaptic labeling (*Figure 4c, d*) was performed in eight cortical sections from one mouse. Representative examples were selected from the sets of bio-logical replicates.

All Triton control sections, used for comparison with the permeabilization-free condition (*Figures 1* and *3–5*), were acquired from either adjacent sections or the opposite hemisphere of the same animal. The HRP experiment illustrated in *Figure 5i* was performed with two mice, each with four sections. The proof-of-principle correlative IHC and EM experiments in *Figure 6* were each per-formed once. The data presented in Supplemental Figures were collected with the following number of mice (N) and number of sections (n): *Figure 1—figure supplement 2* (N = 2, n = 20), *Figure 1—figure supplement 3a* (N = 1, n = 9), *Figure 1—figure supplement 3b* (N = 1, n = 6), *Figure 1—fig-ure supplement 4* (N = 2, n = 8), *Figure 1—figure supplement 5* (N = 1, n = 6), and *Figure 1—fig-ure supplement 6* (N = 1, n = 20). Micrographs from sections treated with SeeDB in *Figure 1—figure supplement 2* are shown again in *Figure 1—figure supplement 6* to facilitate a comparison of ultrastructural quality with and without SeeDB.

## Co-localization analysis

For the co-localization analysis of synaptophysin (*Figure 4d*), we used a method similar to a previ-ously described rotation-based strategy (*Soto et al., 2011*). Two simultaneously acquired fluorescent 2P stacks represented the labeling of synaptophysin without Triton (AF488, green channel) and sub-sequently with Triton (AF594, red channel). We analyzed whether presynaptic puncta were labeled with Triton permeabilization (AF594, red channel) that had not been previously labeled without Tri-ton permeabilization (AF488, green channel). Presynaptic puncta were manually outlined with 0.4-μ m-diameter ROIs (regions of interest), and the voxel intensities were averaged within each ROI in the (AF594) red channel. These ROI intensities were plotted against the same ROI intensities in the (AF488) green channel to assess the co-localization of the synaptophysin labeling before and after Triton permeabilization. We then compared this correlation to a 90° rotation of the (AF488) green channel. A linear regression t-test was used to test the statistical significance of the co-localized cor-relation compared to the rotated correlation.

### Alignment of 2P data to EM data

For the alignment of 2P stacks to EM data (*Figure 6*), we first manually identified 10–20 control points based on matching features such as blood vessels or somata centers. An affine transformation between the control points was fit using Matlab. This transform was then applied to the 2P stacks to register them to the EM data.

## Acknowledgements

We thank B Fubara for assistance with animal perfusions and M Pallotto for assistance with virus injections. We also thank J Diamond, S Haverkamp, and M Pallotto for comments on the manuscript.

## Additional information

### Funding

| Funder | Grant reference number | Author |
| --- | --- | --- |
| NINDS | NS003133 | Kara A Fulton<br>Kevin L Briggman |
| Stiftung Caesar | | Kara A Fulton<br>Kevin L Briggman |

The funders had no role in study design, data collection and interpretation, or the decision to submit the work for publication.

### Author contributions

Kara A Fulton, Conceptualization, Formal analysis, Investigation, Methodology, Writing - original draft, Writing - review and editing; Kevin L Briggman, Conceptualization, Formal analysis, Supervision, Funding acquisition, Writing - original draft, Writing - review and editing

### Author ORCIDs

Kara A Fulton https://orcid.org/0000-0002-9987-2815
Kevin L Briggman https://orcid.org/0000-0003-2946-4661

### Ethics

Animal experimentation: All animal procedures were conducted in accordance with US National Institutes of Health guidelines, as approved by the National Institute of Neurological Disorders and Stroke Animal Care and Use Committee (ASP 1340).

### Decision letter and Author response

Decision letter https://doi.org/10.7554/eLife.63392.sa1
Author response https://doi.org/10.7554/eLife.63392.sa2

## Additional files

### Supplementary files

• Supplementary file 1. Optimized permeabilization-free immunohistochemical (IHC) protocol. Incubation solutions, durations, and temperatures are reported along with reference to relevant supplementary figures. The duration of antibody incubation was varied dependent on section thickness. Asterisk (*) indicates sodium phosphate buffer (PB) concentration should be varied to achieve the desired extracellular space (ECS) volume fraction in a tissue-dependent manner. ACSF: artificial cerebral spinal fluid; PFA: paraformaldehyde; GA: glutaraldehyde; Ab: antibody; CB: sodium cacodylate buffer; NIRB: near-infrared branding; RT: room temperature; o/n: overnight.

• Transparent reporting form

## Data availability

All data analysed during this study are included in the manuscript and supporting files. The 3D SBEM volume in Figure 6 is available for online browsing and annotation at https://wklink.org/1836 as detailed in the data availability section of the Methods.

The following dataset was generated:

| Author(s) | Year | Dataset title | Dataset URL | Database and Identifier |
|---|---|---|---|---|
| Fulton KA, Briggman KL | 2021 | K0068_05, mPFC SBEM volume | https://wklink.org/1836 | webknossos, wklink.org/1836 |

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
