## [Decision Letter]

Thank you for submitting your article "Permeabilization-free en bloc immunohistochemistry for correlative microscopy" for consideration by *eLife*. Your article has been reviewed by 3 peer reviewers, including Albert Cardona as the Reviewing Editor and Reviewer #1, and the evaluation has been overseen by John Huguenard as the Senior Editor. The following individual involved in review of your submission has agreed to reveal their identity: Christel Genoud (Reviewer #3).

The reviewers have discussed the reviews with one another and the Reviewing Editor has drafted this decision to help you prepare a revised submission.

Summary:

The paper by Fulton and Briggman shows how subtle alterations in the way that brain tissue is chemically fixed, altering the amount of space between cells and their processes, increases the penetration depth of antibodies. This allows different epitopes to be revealed hundreds of microns into the tissue, but without the use of detergents, or chemicals that would disrupt the ultrastructure for analysis with electron microscopy. The work is thorough and complete, and boasts a refreshingly honest and straightforward discussion on the strengths and limitations of the method, and its potential for future application.

This manuscript describes exhaustively and with all appropriate controls a new method to perform immunohistochemistry for fluorescence microscopy in thick tissue (up to 1 mm) by keeping extracellular space, without permeabilization and by preserving the ultrastructure of the mouse brain tissue. The authors show how the fluorescent label can be seen through the entire thickness of fixed sections of, as tested, up to 1 millimeter in thickness, and correlating these images with those from the same volume taken with block face scanning electron microscopy allows the observer to study the same structures with the two imaging modalities.

The issue of compromising the quality of tissue preservation so that antibodies can pick out molecules in the ultrastructure has dogged neuroscience researchers for a very long time. The method described could be a game changer to solve many scientific questions that could not be addressed so far due to the impossibility to keep ultrastructure good enough to reconstruct all neurons while at the same time having en-bloc immunostaining to localize key molecular components through the same volume.

In order to image the fluorescence signal through all the thickness of the tissue, a step of clearing has been added, which as demonstrated, does not compromise the ultrastructure of the tissue. Such a tissue preservation allows to perform serial-block-face SEM on the same piece of tissue and reconstruct neuronal arbors. By merging the images of the 2 modalities, authors are offering the potential to add molecular characterization to the ultrastructural information of a neuronal connectome.

The quality of the data is outstanding and allows reader to judge the impact of all the documented steps. Furthermore, the care to experiment and document many parameters makes it a reference article for future researchers optimizing immunohistochemistry on thick samples and clearing techniques. Text is focused, clear and perfectly organised.

However, the paper is less impressive as a method for large volume analysis. The figures 1 to 5 clearly describe the ECS method and provide an excellent description of how this is carried out. Multiple examples of staining through 300 µm of tissue are possible using a range of labels.

The examples used to illustrate the strengths of the method in Figure 6 fall short of being the most convincingly possible to demonstrate the power and novelty of the technique. The first example relies on GFP, a genetically encoded protein, while one strength of the method is that you can label molecules that are not/cannot be genetically labelled.

Studies have been published that do not need to preserve fluorescence (registration by blood vessel and cell bodies [Maclachlan et al. 2018 Front Neuroanat; https://doi.org/10.3389/fnana.2018.00088 ]), where there's no need to perform DAB due to near-infrared branding fiducials. But admittedly these aren't in thick tissue sections.

In addition, in Figure 6 where the correlative light-EM (CLEM) is demonstrated, examples are shown where it could be argued that the ECS method is unnecessary. In the first demonstration the authors reveal eGFP with immunocytochemistry inside a section from the olfactory bulb. It's not clear if this neurite was first selected in the slice from the light microscopy images, and then localized in the stack of EM images, or simply selected post hoc, once the tissue was embedded and cut. Therefore, its not clear what this adds to this paper describing a correlative method.

The second example in Figure 6 with anti-TH is more relevant but still consists in the labeling of entire axons rather than for example a synaptic marker, which would demonstrate the enriching of EM-connectome information with molecular detail. Furthermore, here it's not shown how deep this labelled axon is in the slice, and although its described as being 300 µm thick the volume imaged appears to be only 100 µm. What is more, the authors are targeting a large axon with clear morphology. This raises the question as to whether or not it would even be possible to find some of the smaller fluorescent labeling shown in the previous figures, and at a significant depth. For example, could the Homer labeling be correlated with features in serial EM images deep inside a tissue section?

While the reviewers agree that further experiments are most likely not necessary, we ask for a significant revision of figure 6, perhaps with additional data that the authors already have, including details on how were the samples selected and matched across imaging modalities, as well as including CLEM demonstrations on the thicker, up-to-1 mm samples, particularly for non-genetically encoded markers.

Essential revisions:

Please revise Figure 6 and associated text as suggested.

---

## [Author Response]

Essential revisions:Please revise Figure 6 and associated text as suggested.In addition, in Figure 6 where the correlative light-EM (CLEM) is demonstrated, examples are shown where it could be argued that the ECS method is unnecessary. In the first demonstration the authors reveal eGFP with immunocytochemistry inside a section from the olfactory bulb. It's not clear if this neurite was first selected in the slice from the light microscopy images, and then localized in the stack of EM images, or simply selected post hoc, once the tissue was embedded and cut. Therefore, its not clear what this adds to this paper describing a correlative method.

We agree that this figure is not the most relevant demonstration of CLEM. Our intention was to demonstrate that, in addition to fluorophore-conjugated secondary antibodies, a larger enzyme like HRP could be used as a reporter and remain compatible with the permeabilization-free protocol. This is an alternative to, for example, the expression of a genetically encoded peroxidase such as APEX-2. We have moved the GFP/HRP panels to Figure 5 and modified the associated text in lines 185 – 190 and the Figure 5 legend.

The second example in Figure 6 with anti-TH is more relevant but still consists in the labeling of entire axons rather than for example a synaptic marker, which would demonstrate the enriching of EM-connectome information with molecular detail. Furthermore, here it's not shown how deep this labelled axon is in the slice, and although its described as being 300 µm thick the volume imaged appears to be only 100 µm. What is more, the authors are targeting a large axon with clear morphology. This raises the question as to whether or not it would even be possible to find some of the smaller fluorescent labeling shown in the previous figures, and at a significant depth. For example, could the Homer labeling be correlated with features in serial EM images deep inside a tissue section?While the reviewers agree that further experiments are most likely not necessary, we ask for a significant revision of figure 6, perhaps with additional data that the authors already have, including details on how were the samples selected and matched across imaging modalities, as well as including CLEM demonstrations on the thicker, up-to-1 mm samples, particularly for non-genetically encoded markers.

To address the reviewers’ concerns about the CLEM demonstrations, we have added additional data to Figure 6, panels a and b. The new data demonstrates the triple immuno-labeling of calretinin (CR), calbindin (CB), and tyrosine hydroxylase (TH) expressing interneurons throughout a 300 µm thick section of the mouse olfactory bulb. We then processed the same section for EM and cut ultrathin sections until reaching approximately the middle of the section depth (~150 µm). We identified the corresponding immuno-labelled somas in the two-photon stack and a 35 nm EM section. We also provided additional details on how the alignment of the fluorescent stack and EM data was performed in the Methods section. We believe this addresses the reviewers concerns about the ability to correlate permeabilization-free labeling of proteins in large-volume samples. The new text describing this experiment can be found in lines 201 – 210 and in Methods sub-section ‘Triple labelling of olfactory bulb’, lines 411 – 420.

For the CLEM experiment labeling TH^+^ axons, we adjusted Figure 6, panel c, to show the full depth of the 300 µm thick section and the depth of the 3D SBEM volume with the section. We revised the relevant text and reported the depth of the axons traced in lines 217 – 227.